# Body Size, Not Personality, Explains Both Male Mating Success and Sexual Cannibalism in a Widow Spider

**DOI:** 10.3390/biology10030189

**Published:** 2021-03-03

**Authors:** Rok Golobinek, Matjaž Gregorič, Simona Kralj-Fišer

**Affiliations:** Jovan Hadži Institute of Biology, Research Centre of the Slovenian Academy of Sciences and Arts, Novi trg 2, 1000 Ljubljana, Slovenia; rok.golobinek@gmail.com (R.G.); matjaz.gregoric@zrc-sazu.si (M.G.)

**Keywords:** *Latrodectus tredecimguttatus*, sexual conflict, aggression, invertebrate personality, oral sexual contact, sexual size dimorphism

## Abstract

**Simple Summary:**

The role of personality in sexual selection has mostly been investigated on vertebrate species, in which males provide direct benefits to females or offspring. Less is known about the links between behavioral variation and sexual selection in species where males provide only sperm, while the advantages of female-choice are due to male genes that increase offspring fitness. Our study is centered on a sexually-size dimorphic spider species, the Mediterranean black widow, which is ideal to investigate how sexual selection shapes behavior. In this species, a male-biased operational sex-ratio leads to male-male competition, and aggressive and/or large males should have a selective advantage. Females are selected for fecundity, which should correlate with selection for higher voraciousness. Theory predicts that voracity “spills-over” into the mating context, such that voracity towards prey correlates with voracity towards mates. We tested how body size and two behaviors, male aggression toward rivals and female voracity toward prey, influence mating behavior, mating success, and sexual cannibalism. We show that individual variation in aggression does not play a direct role in the mating behavior of this species. Instead, body size affects male mating success and the occurrence of sexual cannibalism in females.

**Abstract:**

Theory suggests that consistent individual variation in behavior relates to fitness, but few studies have empirically examined the role of personalities in mate choice, male-male competition and reproductive success. We observed the Mediterranean black widow, *Latrodectus tredecimguttatus*, in the individual and mating context, to test how body size measures and two functionally important aggressive behaviors, i.e., male aggression towards rivals and female voracity towards prey, affect mating behaviors, mating success and sexual cannibalism. We specifically selected voracity towards prey in females to test the “aggressive spillover hypothesis”, suggesting that more voracious females are more sexually cannibalistic. Both females and males exhibit consistent individual differences in the examined aggressive behaviors. While larger males win contests more often and achieve more copulations, neither male nor female size measures correlate to aggression. Female voracity does not correlate with aggression towards mates and sexual cannibalism, rejecting the “spillover hypothesis”. However, occurrence of sexual cannibalism positively relates to longer insertion duration. Furthermore, the smaller the ratio between male and female body length the more likely a female attacked and cannibalized a mate. We show that individual variation in aggression levels plays no direct role in the mating behavior of the Mediterranean black widow. Instead, body size affects male mating success and occurrences of sexual cannibalism in females.

## 1. Introduction

Animal personalities, or consistent individual differences in behavior that persist through time and across contexts, have been studied intensively in the last decades [1,2], and occur in many taxa, including invertebrates [3]. Personalities have complex effects on fitness and have been suggested to entail ecological and evolutionary implications [4]. Nevertheless, few studies have empirically tested for the relationships between individual variability in behavior and fitness consequences, and those show mixed results (reviewed in [5]).

A likely explanation for the variation in the results of past studies is that fitness effects of a behavioral type are often context, condition, and/or sex-dependent [6,7,8,9]. For example, high activity can help males encounter more females and high aggression can increase their success when competing for access to females, both enhancing reproductive success. However, such attributes may be costly and disadvantageous in an antipredator context, where active and aggressive males may suffer higher survival costs [10]. A well-known example of such a trade-off across contexts is a model called the aggressive spillover hypothesis, which was proposed to explain sexual cannibalism in the raft spider *Dolomedes fimbriatus* [11]. This model suggests that juvenile female aggression towards prey was selected to increase growth rate and fecundity, but “spills over” into aggression towards courting males in adulthood due to genetic constraints. Consequently, highly voracious females attack and kill approaching males and thereby remain unmated [11].

The aggressive spillover hypothesis has been discussed frequently, but most studies support it only partly [12]. Namely, several studies show a correlation between female voracity and precopulatory cannibalism, e.g., in *Dolomedes triton* [13], *Argiope aurantia* [14], and *Lycosa hispanica* [15], but the evidence for maladaptiveness of high female voracity and sexual cannibalism are lacking (reviewed in [12]) or studies find effects in the opposite direction than predicted [16]. Furthermore, a follow up study on *Dolomedes fimbriatus* found no support for most of the predictions of the aggressive spillover hypothesis. While females exhibited consistent individual differences in voracity towards prey, voracity did not correlate with individual variability in aggression towards males and sexual cannibalism [16]. However, both female aggression towards males and sexual cannibalism correlated with the relative size difference between mates, where females more likely consumed relatively smaller males [16]. The latter indicates sexual cannibalism as a mechanism of mate choice based on male size, a pattern found in some other spider species (e.g., [17,18]).

While sexual selection studies mostly focus on the effects of body size or other morphological traits (color, ornaments), behavioral traits may also play an important role in intra-sexual competition and mate choice [19]. The links between behavioral variation and sexual selection have been mainly investigated in vertebrate systems [8,19]. For example, in great tits (*Parus major*) and zebra finches (*Taeniopygia guttata*), behavioral traits are involved in mate choices of both sexes. Similarly, older males of the banded wren (*Thryothorus pleurostictus*) sing more consistent songs and are more likely to mate than younger males [20]. Thus, consistency is another source of potential information in sexual selection contexts, and sexual selection, therefore, likely affects the maintenance of personality traits [19].

In spiders, the importance of behavioral traits in male–male competition and mate choice is understudied. For instance, in the bridge spider (*Larinioides sclopetarius*)*,* aggression in males correlates positively to the number of sired offspring, implying that aggressive males have an advantage in sperm competition [21]. The significance of male aggression in mate choice has been shown in the hermit spider *Nephilingis livida*, where more aggressive males are less likely to be cannibalized by females, and also achieve higher copulation rates [22]. In *D. fimbriatus,* however, there is no support for the effects of such among-individual differences in behavior on mating success and sexual cannibalism [16].

Here, we explore the relationship among individual variation in behaviors, body size measures and mating success in the Mediterranean black widow (*Latrodectus tredecimguttatus*). Characteristically for widow spiders (genus *Latrodectus*), this species exhibits sexual size dimorphism (SSD) with females several-fold larger than males. Males of *L. tredecimguttatus* are often monogamous, though not obligatorily [23], while females are presumed to be polygamous [23]. Several mating behaviors associated with SSD are present in this species, including sexual cannibalism, male genital damage, and female genital plugging and mate guarding [24]. These extreme mating behaviors and SSD indicate widow spiders are under strong sexual selection [25], which makes them an appropriate study group to explore the role of behavioral traits in the mating context.

We observed the Mediterranean black widow in the individual and mating context, to test how body size measures and two functionally important behaviors, i.e., male aggression towards rivals and female voracity towards prey, affect mating behaviors, mating success, and sexual cannibalism. Specifically, we asked (i) whether males and females consistently differ in aggression towards a rival and voracity towards prey, respectively; (ii) whether and how behavioral traits correlate to measures of body size in both sexes; (iii) whether and how male aggression and female voracity, together with size measures, affect the behavior of individuals in the mating contexts; and (iv) whether size and behaviors correlate with male mating success (number and duration of copulations) and sexual cannibalism.

To broaden our understanding of extreme sexual behaviors connected to SSD, we additionally quantify mating behaviors in *L. tredecimguttatus* in detail. We predicted that both sexes exhibit consistent individual variation in aggressive behavior. If increased aggression correlates to increased foraging success, we expected to find a positive correlation between aggression and body size measures. We predict larger males will win contests more often, and thus be more likely to gain mating opportunities. Furthermore, aggression may be beneficial for mating success, in particular in large males. Aggression in females, however, may limit their mating success if it spills over into aggression towards mates. In females, large body size advertises high fecundity [26,27], and we thus predicted that larger females are more attractive to males and achieve higher mating success (number and duration of copulations). Furthermore, we tested two additional hypotheses for sexual cannibalism. The mate-size dimorphism hypothesis predicts that sexual cannibalism depends on relative size difference between the male and the female [17,28]. Sexual cannibalism may also allow females to control copulation duration and thereby paternity of males [29,30]. Under assumptions that females should benefit from mating with more than one male, the probability of sexual cannibalism should increase with duration of copulation.

## 2. Material and Methods

### 2.1. Study Animals

We collected egg sacs of *L. tredecimguttatus* in the field around Pula, Croatia (N44.940882°, E13.8718884°) at the beginning of their hatching season in early May 2017. We kept individual egg sacs in plastic containers until hatching. After their second molt, we moved individual spiders into their own plastic containers (5 cm × 10 cm) with a foam cover. From the initial egg sacs, we reared two generations of spiders, which we used for this study. We kept all spiders in standardized laboratory conditions, at 21–24 °C, 40–60% relative humidity, and natural lighting conditions. We sprayed and fed spiders twice per week, using size-appropriate food: *Drosophila melanogaster* for small juveniles and males, *Lucilia sericata* for middle sized juveniles and larvae of *Tenebrio molitor* for adult females. To facilitate web construction for later experiments, we moved penultimate females onto wooden or metal constructions (20 cm × 12 cm × 11 cm), consisting of a horizontal plate and six vertical sticks (Appendix A). The feeding and spraying regimes remained the same for spiders on these constructions. We recorded each experiment using a Panasonic HC-V180 camcorder, and we conducted all behavioral quantifications from recordings.

### 2.2. Female Voracity towards Prey

We quantified female voracity as her reaction to prey. Each female (N = 54) was tested for voracity twice before being in the mating experiment. After reaching adulthood, all females were fed with one food item, three days later tested for voracity and allowed to eat the prey, and then tested again for voracity three days later, then finally put into a mating trial after another three-day period. Before each experiment, we set the female’s web in front of a black background and between lights to visualize the web and spider for video recording. These spiders are active during the day and night, so the lights likely did not affect results. After 15 min of acclimatization, we placed a mealworm larva on the edge of the enclosure and allowed it to roam between sticky threads of the web (Appendix A). We started quantification when prey first touched the web, and ended once the spider finished manipulating it, i.e., ignored it for at least 45 s or started feeding. We scored the following behaviors.

Latency to first reaction: time from web contact of prey to the first movement of the spider.Latency to wrap prey: time from web contact of prey to beginning of prey wrapping.Latency to bite prey: time from web contact of prey to first biting of prey.Prey manipulation time: time from first reaction to the end of the experiment.

### 2.3. Aggression in Males

We quantified a males’ aggression as his behavior towards a conspecific male. Each male (N = 59) was tested twice, once on an empty adult female web and once in a mating trial. Female webs were chosen at random. In the first experiments, we randomly selected two males and placed them on an empty female web (Appendix A). Males were paired with a different opponent in each test. Because most males matured within 1 and 2 weeks of each other, male age was ignored when pairing for personality experiments. To distinguish the males, we marked each on one of the legs using nail polish of different colors. We measured several antagonistic behaviors during a period of one hour or until one of the males lost the contest (escaped or fell from the web and stayed off it for more than 30 s). If neither male fled (N = 6), we declared the spider that was closest to the female retreat after one hour, the winner. Males started at opposing ends of the web, so they could meet as naturally as possible. Because males were highly aggressive during the first contest, with several losing a leg, we conducted the second contest as part of the mating trial to prevent larger injuries before mating trials. This contest again concluded if one of the males fell or escaped from the web (and stayed off for more than 30 s), or additionally, if one of the males mated with the female (we considered the mating as a win). If neither male mated or escaped, we again declared the male closest to the female retreat as the winner after 1 h. We immediately removed the losing male, to prevent him from interfering with the mating, in order to know the paternity of offspring for a related study. Males injured in the first experiments were not disqualified from mating experiments, because the loss of a leg did not appear to hinder their behavior. We scored the following behaviors:Latency to first attack: time from the beginning of the experiment to the first attack. We defined an attack as any rapid movement towards the opponent in close proximity (see Appendix A).Number of attacks: total number of attacks that a male conducted towards its opponent. Scored separately for each male.Number of fights: total number of fights. A fight was defined as an attack that resulted in physical contact of the competing males.Total fight duration: the summed duration (s) of all fights. We defined fight duration as time from attack until the escape or death of one of the spiders.Number of movements towards opponent: total number of times that a male responded to the presence of its opponent by stopping his courting behavior and walking towards the opponent, occasionally shaking the web. Chance encounters were not included, and were defined as random encounters between two courting males.Number of chases: total number of times the male chased his fleeing opponent. We defined a chase as the immediate movement of a male towards a fleeing opponent.Aggressive web shaking: number of times the male shook the web at his opponent. Since males also use web shaking for signaling to the female, this was only counted when males shook the web near a rival male, not during courtship.First attacker: whichever male attacked first, measured as binary, yes/no.Number of escapes: total number of times the male escaped fighting during the experiment. An escape was defined as rapid movement away from the opponent.Winner: whichever male won the contest (criteria defined above). Measured as binary, yes/no.Experiment duration: total time of experiment.

### 2.4. Mating Trials

During mating trials (N = 30), we scored mating and antagonistic behaviors of both males, while competing for the female, as the second contest to quantify male aggression, and continued quantifying the winner’s behavior after removal of the losing male. Mating experiments were done as soon as possible, following the second personality experiment of the female (within 3 weeks of female maturation). If the female killed the male during copulation (N = 9), the experiment was considered over only after the males’ pedipalp was detached from her genitals. If the male survived mating and continued courting, we ended the experiment after a total of three hours, unless the male was currently mating at the three-hour mark. In this case, the experiment was extended until the end of mating (N = 1). We scored the following mating behaviors:First contact: time from start of experiment to the first physical contact between the male and the female.Silk deposition: male deposition of silk on the female web. Measured binary (yes/no for occurrence of the behavior) in intervals of one minute, for each male separately.Web reduction: a male collapsing some of the female’s web using his chelicerae. Measured binary (yes/no for occurrence of the behavior) in intervals of one minute, for each male separately.Mate binding: male silk deposition on the female’s body. Measured binary (yes/no for occurrence of the behavior) in intervals of one minute, for each male separately.Oral sexual contact: contact between male chelicerae and the female’s genital opening. Measured binary (yes/no for occurrence of the behavior) in intervals of one minute, for each male separately.Latency to first insertion: time from beginning of experiment to insertion of the male’s palp into the female’s genital opening.Inserted palp: which palp was inserted (left/right).Insertion duration: total time of insertion for each palp separately.Sexual aggression: measured binary (yes/no) for whether the female attempted to bite the male, grasped at him with her legs, or attempted to cover him in silk.Sexual cannibalism: measured binary (yes/no) for whether the female killed the male during mating.Total insertions: total number of pedipalp insertions.Total insertion duration: summed time of all pedipalp insertions.

All behaviors were measured as frequencies, meaning that uninterrupted behavior was not counted more than once. For instance, a male shaking the female’s web for 10 s was only counted as shaking it once, while a male that stopped and moved between shakings had the behavior counted each time he started.

After noticing web reduction and silk deposition behavior by males, we conducted a subsequent experiment, testing for pheromones in female silk. We collected entire female webs on a sterile inoculation loop. Following [31], we then washed the collected silk in methanol to extract potential pheromones. We soaked sterile filter paper in 50 µL of silk extract, and put each filter paper on a T-shaped climbing structure [32]. We covered each filter paper with a plastic cage to prevent males from contacting it. We presented 10 males each to their own such silk extract and tested whether they will locate the filter paper and start courtship.

### 2.5. Body Size Measurements

Before all experiments, we photographed each spider using standardized settings on a Canon EOS 7D DSRL camera equipped with a 50 mm Canon macro lens. We used these photographs to later measure body length (prosoma + opisthosoma), as a linear measure of body size [33]. Because removing females from their silken retreats caused significant web destruction, we used abdomen volume as a proxy of their body mass [34,35]. We measured abdomen volume using the formula $V=\frac{3}{4}\cdot \pi \cdot \frac{a\cdot b\cdot c}{2}$ where a = abdomen length, b = abdomen width, and c = abdomen height [34,35]. We acquired abdomen measurements from photographs obtained by the above protocol. In addition, we calculated body condition in females. We used standardized residuals of abdomen volume over body length using linear regression.

### 2.6. Ethical Note

Research on spiders is not restricted by the animal welfare law of the country where the study was conducted. In the field, we collected the minimum number of individuals needed to conduct the research. The spiders were kept in conditions similar to their natural environmental conditions. The spiders were regularly fed with different prey items. The study was based on behavioural observations and all experiments were non-invasive. After the experiments, the spiders remained in the laboratory and were reared until natural death as described above.

### 2.7. Statistical Analyses

We analyzed whether females and males differ consistently in behaviors related to voracity towards prey and antagonistic behaviors towards rivals, respectively. Due to the large number of observed behaviors, we first reduced them by performing principal component analysis (PCA) with varimax rotation. In females, latency to first reaction, latency to wrap prey, and latency to bite prey were included in PCA, while duration of prey manipulation was analyzed separately, because it was not correlated to the former three behaviors. In males, we included in the PCA number of approaching the rival, number of web shakings, number of chases, attack latency, number of attacks, number of fights and fight duration. Because the two remaining behaviors, the attack first and winner-loser, are binary, we analyzed them separately. To estimate the relevant number of factors to extract, we used Kaiser’s eigenvalue-one rule and the screen test. To estimate an individual’s scoring on the extracted factor, as an underlying measure for a suite of correlated behaviors, we used the Anderson–Rubin method. The PCA determinants of the correlation matrices exceeded 0.0001, confirming there was no multicollinearity or singularity in the data subjected to PCAs. Measures of sampling adequacy indicated that the correlation matrices were appropriate for PCAs (Kaiser–Meyer–Olkin measure of sampling adequacy: >0.5, Bartlett’s test of sphericity: *p* < 0.05).

We then used generalized mixed models (glmm) to analyze individual repeatability of behaviors. To partition phenotypic variance into between-individual (Vind) and within- individual variance (Ve), we ran Markov Chain Monte Carlo (MCMC) glmm in R [36], following [37]. We calculated repeatability as $r\;=\frac{Vind}{Vind+Ve}$ [38]. In the models, we used trial (repeat 1, repeat 2) as a fixed factor, and animal ID as a random factor. However, when calculating the repeatability of male aggression, we added the ID of the contest as a random factor to account for the effect of an opponent’s behavior. We chose this approach because an individual’s aggression in the contest may depend in part on his rival’s behavior. We ran models using non informative priors. We checked convergence and mixing properties by visual inspection of the chains and checked the autocorrelation values. To verify that the number of iterations was adequate for MCMC chains to achieve convergence, we ran Heidelberger and Welch’s convergence diagnostics.

Next, we tested for the relationships between body size measures (body length, abdomen volume, and body condition) and behaviors that were significantly repeatable, i.e., voracity towards prey (primary factor from PCA analysis, see Results) and duration of prey manipulation in females, and with escalated aggression (aggression that continues beyond just a confrontation, e.g., fighting; also see Section 3.2) and winner–loser in males. In females, we used Pearson’s correlation coefficient to test for correlations. We used the means of behaviors across the two repeats, and mean voracity was log transformed. Then, we tested whether an individual’s behavior within the mating context correlates to female aggressive behavior and body size measures (body length, abdomen volume). Using linear regression (continuous variables) and logistic regression (binary variables), we tested whether and how behaviors (voracity towards prey and duration of prey manipulation) and body measures affect mating success (number of copulations, total duration of copulations), aggression towards mates (yes/no), and sexual cannibalism (yes/no). Regressions were run with bootstrapping. In addition, we tested whether female aggression and sexual cannibalism depended on total copulation duration and the relative size difference between a male and female in a pair (males’ length: female length) using regression.

In males, we used MCMCglmm to test for the relationship between male aggressive behaviors (escalated aggression), winning the contest and body size measures. We asked whether and how male aggressive behaviors (escalated aggression), body length and their interaction (aggression × body length) affect winning the contest. We ran this model because we needed to account for an opponent’s behavior, thus we added experiment ID as a random variable.

Finally, we ran linear (continuous variables) or logistic regression (binary variables) to test how male body length, aggression, and courting activity affect mating success (yes, no), number of copulations (i.e., pedipalp insertions), total duration of copulations, received female aggression (yes/no), and occurrences of being cannibalized (yes/no). Regressions were run with bootstrapping.

We reduced the number of the recorded courtship behaviors (occurrences of silk deposition, web reduction, mate binding and oral contact) using PCA, as described above.

For the descriptions of mating behaviors, we analyzed only winning males from the mating contests, as losing males did not have a chance to fully court females. We report descriptive statistics as medians ± interquartile ranges.

## 3. Results

### 3.1. Female Voracity towards Prey

Principal components analysis (PCA) identified one primary factor with Eigenvalues greater than 1, which explained 66.38% of total variance. We named this factor “voracity towards prey”. Loadings higher than 0.5 were found for all behaviors subjected to PCA: latency to the first reaction, latency to wrap prey, and latency to bite prey. Females were more voracious in the second compared to the first trial, but this difference was not statistically significant (post. mean difference = −7.944, 95% credible interval (CI) = [−17.045, 1.392], *p* = 0.087). We analyzed the duration of prey manipulation separately, and it did not differ between the two repeated trials (post mean difference = 49.98, 95% CI = [−48.06, 148.08], *p* = 0.312).

Females exhibited consistent individual differences in both voracity towards prey and duration of prey manipulation (Table 1). Neither body length, abdomen volume nor body condition correlated to voracity towards prey or duration of prey manipulation (Table 2).

### 3.2. Aggression in Males

PCA suggested two primary factors with Eigenvalues greater than 1. Factor 1 and factor 2 explained 48.25% and 15.36% of total variance, respectively. Factor 1, which we named “aggression”, had loadings higher than 0.5 for number of approaching the rival, number of web shakings, number of chases, number of attacks and number of fights. We named the second factor “escalated aggression”, which had significant loadings for attack latency and fight durations. Because they are binary, we analyzed the behaviors “winner-loser” and “attack first” separately. Behavior levels did not differ between the two trials (aggression: post mean difference = −0.023, 95% CI = [−0.428, 0.386], *p* = 0.922; escalated aggression: post mean difference = 0.064, 95% CI = [−0.416, 0.518], *p* = 0.789; attack first: post. mean difference = 0.283, 95% CI = [−0.403, 1.008], *p* = 0.434; winner-loser: post mean difference = 0.254, 95% CI = [−0.482, 0.970], *p* = 0.493).

Males exhibited consistent individual differences in escalated aggression and winning or losing the contest, whereas estimated repeatability of aggression and attacking first was low (Table 1). While winning versus losing fights was related to body size, where winners were larger (post mean = 0.026, 95% CI = [< −0.001, 0.054], *p* = 0.038), escalated aggression was independent of size (post mean = <−0.001, 95% CI = [−0.003, 0.002], *p* = 0.759).

### 3.3. Relationships among Personality, Body Size, and Mating Behaviours

We found no relationship between female traits (body length, abdomen volume, voracity towards prey or duration of prey manipulation) and their copulation success (number of copulations, total duration of copulations; Table 3). Sexual cannibalism was related to female abdomen volume. Namely, smaller females more often cannibalized their mates (B = −0.028, Bias = −4.479, SE = 139.719; 95% CI [−0.168, 0.003], *p* = 0.050; Table 4). Neither female aggression towards males nor sexual cannibalism related significantly to any of other female traits (Table 4). Female aggression towards a mate and sexual cannibalism however related to the ratio between male and female body length (male: female); the smaller the ratio the more likely a female attacked and cannibalized a mate (Table 4; Figure 1).

Males that were larger won contests significantly more often than smaller males (post mean difference = 0.026, 95% CI = [<−0.001, 0.054], *p* = 0.038). Neither aggression nor interaction between aggression × body size affected the contest outcome (results of the full model in Table 5; Figure 2 and Figure 3).

Twenty out of 30 winners (66.7%) mated with the female. Mating success (no/yes) of the winner however was not dependent on their courting activity, aggression, and body length (Table 6). Larger males achieved more pedipalp insertions, but the result was not statistically significant (B = 0.327, Bias = 0.022, SE = 0.199; 95% CI [−0.028, 0.754], *p* = 0.090; Table 6). Courting activity, body length and aggression did not affect the number of copulation insertion (Table 6). Female aggression towards males and sexual cannibalism was unrelated to male body length, courting activity and aggression (Table 7).

### 3.4. Mating Behaviours

During mating experiments, females mostly remained inactive in their retreats. While courting, males deposited silk on the female’s web, engaged in female web reduction, web shaking, mate binding, and oral sexual contact (Appendix A).

Males began depositing silk immediately on contact with the female web and alternated this with the reduction of the female’s web. Males that mated (N = 20) engaged in 3–92 (μ_1/2_ = 23.5 ± 27.25) silk deposition and 2–58 (μ_1/2_ = 20 ± 39) web reduction events before first pedipalp insertion. All males (N = 10) in the subsequent pheromone trial, located the filter paper despite being prevented from direct contact, and started depositing their own silk onto the plastic cage. Males approached females slowly, often shaking their webs. They shook their webs also when retreating after female aggression. After first contact with the female (after 3.92–50.68 min, μ_1/2_ = 11.72 ± 28.61 min), the male climbed onto the dorsal side of the female’s abdomen, and began depositing silk onto the female, i.e., engaged in mate binding. Males engaged in 1-33 (μ_1/2_ = 14.5 ± 20.25) mate binding events before first pedipalp insertion. Along with mate binding, males routinely engaged in oral sexual contact, i.e., approached the female’s genitals and salivated into her copulatory openings (2–38, μ_1/2_ = 15.5 ± 17.75). High frequency drumming of the male’s abdomen on the female’s abdomen accompanied both oral contact and mate binding, and this behavior occurred even when the male was standing idle on the female’s abdomen. Males achieved first insertion after 12.18–162 min (μ_1/2_ = 61.89 ± 68.47 min), which lasted for 0.33–40.47 min (μ_1/2_ = 12.83 ± 19.47 min). Males inserted a palp once in 55%, twice in 35%, three times in 5%, and four times in 5% of matings. After first insertion, 25% of males continued with silk deposition and web reduction, while 50% of males stayed on the female and continued with mate binding and oral sexual contact. The second insertions lasted for 1.15–38.45 min (μ_1/2_ = 9.5 ± 17.08 min, N = 9), and did not differ from the first in length (*p* = 0.757, Man–Whitney *U* = 83). 25% of males were cannibalized during their first pedipalp insertion, and 44% of males that copulated a second time were cannibalized. Females were aggressive in 60% of mating experiments, and sexual cannibalism occurred in 45% of experiments. In all experiments with sexual cannibalism, females used it to terminate mating. All cases of cannibalism occurred during pedipalp insertion, and the sexual organs of males remained lodged inside female genitals even while being consumed. Females attacked males more often during first insertion (40% of experiments). Female aggression and sexual cannibalism were more likely when total copulation insertions were longer; however, the relationships were not significant (aggression towards a mate, B = 0.001, SE = 0,001, Wald = 2.575, *p* = 0.109; sexual cannibalism, B = 0.001, SE = 0,001, Wald = 3.455, *p* = 0.063; Figure 4).

## 4. Discussion

### 4.1. Personality Traits and Mating Behaviours

Females of the Mediterranean black widow (*L. tredecimguttatus*) exhibit consistent individual differences in both examined behaviors, voracity towards prey and duration of prey manipulation. Similarly, the individual variation in escalated aggression and winning or losing contests is consistent in males. These results confirm our predictions that these aggressive behaviors are stable personality traits. While larger males win contests more often and achieve higher number of insertions, male body size and mass do not correlate to escalated aggression. In females, body size and mass do not correlate with any of the two behavioral traits and mating success. Female voracity does not correlate with aggression towards mates or sexual cannibalism, rejecting the “spill-over hypothesis”.

Our results on repeatable individual differences in behaviors related to aggression are consistent with previous studies (reviewed in [3,39]), including other spider species, e.g., *D. triton* [13], *L. sclopetarius* [40], *N. livida* [22], *Zygiella x-notata*, and *Nuctenea umbratica* [41]. Furthermore, variation in aggression is moderately heritable (e.g., *L. sclopetarius* [21,40]) and *N. umbratica* ([42], reviewed in [43]) and often has fitness consequences (reviewed in [5]). For example, aggression in *L. sclopetarius* males correlates positively to the number of sired offspring, suggesting that aggressive males have a sperm competition advantage [21]. Here, however, we found no relationship between aggression and mating success as a proxy for fitness consequence.

The aggressive spillover hypothesis suggests that aggression towards prey which is favored in juveniles “spills over” into the mating context due to genetic constraints on the plasticity of aggression [11]. Namely, more aggressive females more likely catch more prey and grow larger, but also consume courting males and may thus remain unmated [11]. While our results show consistent individual differences in female voracity towards prey, we found no correlation between female voracity and aggression towards mates or sexual cannibalism, thus not supporting the aggressive spillover hypothesis in the Mediterranean black widow.

Current studies show mixed results on correlation between voracity towards prey and sexual cannibalism (reviewed in [12]). This might be in part due to sexual cannibalism being affected by other factors, such as hunger, female size and mating status, size dimorphism and mate quality [28]. We show that less heavy females of the Mediterranean black widow are more likely to consume their mates, which is opposite to the outcome proposed by the aggressive spillover hypothesis. However, this result is consistent with findings in other spider species, e.g., *N. plumipes* [44], *D. triton* [45], and *D. fimbriatus* [16].

The female choice hypothesis predicts that sexual cannibalism may allow females to control the duration of copulation [29,30]. Accordingly, our results show that females more often act aggressively and cannibalized males when copulations were longer. This suggests, that females might have controlled the duration of copulation by sexual cannibalism, which would allow them to mate with more than one male (e.g., [22,46]) Under the assumption, that females benefit from mating with more than one male, sexual cannibalism can be viewed as a mechanism to control the paternity share of mates [29,30,47]. The mating system of *L. tredecimguttatus* have not been studied in detail, however it has been reported that males of this species are often monogynous, though not obligatorily [23], while females are likely polyandrous [23]. Sexual cannibalism has also been explained through a male’s advantages. Namely cannibalized males achieve longer copulations and thereby higher sperm transfer and may have more or “better” offspring [48,49,50,51]. Male fitness benefits from sexual cannibalism is apparent in *L. hasselti,* where males self-sacrifice to females during copulation and achieve higher paternity [48]. While sexual cannibalism is common in *Latrodectus* (e.g., [52]), males of other species from the genus including *L. tredecimguttatus* avoid rather than facilitate sexual cannibalism.

Another hypothesis, the mate-size dimorphism hypothesis, predicts that the relative size difference between the male and female predicts the probability of sexual cannibalism [16,17,53,54]. Accordingly, we found that the body length ratio affected the likelihood of a female attack and sexual cannibalism. The smaller the ratio between the female and male, the more likely it was for the female to attack and cannibalize the male.

Consistent individual variation in behavior may be generated by state-dependent positive feedback loops [55,56,57,58] For example, larger individuals may be bolder, more aggressive, and more active than smaller individuals. However, the intrinsic state variables (including body size) explain at best 3–8% of the variation in behavioral traits [59]. In our study, neither male nor female aggression, except for sexual cannibalism (see above), correlated to their body size or mass. Similar results were found in several other studies in vertebrates and invertebrates (e.g., [9,60,61]), including spiders [16,40,62].

We expected to find a positive correlation between male size and mating success (e.g., [63,64,65]) or/and male aggression and mating success (e.g., [22,66]). Our results show that male body size in *L. tredecimguttatus* does not impact whether a male mates or not, however, larger males win male contests more often (e.g., [67]), and tend to achieve higher number of copulations. Other studies have found similar results (e.g., [68,69]). For example, in *Agelenopsis aperta*, females are more likely to mate with larger males, and male mass determines the outcome of male-male agonistic interactions over female [68]. Yet, we found no relationship between male aggression and mating success in *L. tredecimguttatus*, which is in contradiction with the existing evidence on the latter correlation in other species ([21,22,66,70]; but see [71]). Nevertheless, many factors (e.g., size, behavior, mating system, etc.) beyond aggression may simultaneously affect male mating success [9]. In *L. tredecimguttatus,* large size rather than aggression has competitive advantages in winning contests and thus in males’ mating success in staged laboratory trials. Perhaps, large males do not need to be aggressive in male–male competition to achieve good mating success.

### 4.2. Mating Behaviours

The mating strategy in *L. tredecimguttatus* is similar to other congeners. Once a male touches the web of a female, he begins courtship immediately. Web reduction and silk deposition are known in several species, including congeners *L. geometricus* [52] and *L. hesperus* [72]. During web reduction, *L. tredecimguttatus* males cut out sections of the female web and weave them into balls of silk. A similar behavior occurs in *L. hesperus*, where males are hypothesized to remove areas of the female web that are heavily covered in female pheromones [32]. Both male silk deposition and web reduction in *L. hesperus* reduce the attractiveness of the female web for other males [72]. As our pheromone trials indicate the presence of airborne pheromones in female silk of *L. tredecimguttatus*, male web reduction and silk deposition in this species suggests a similar function, i.e., masking the female web to make it less attractive for rival males.

Males approached females carefully, shaking their webs. Web shaking is suggested to calm the female [73]. This is a common signal to the female that the male is a potential partner, not a meal [74], and is in line with our observations, as males also shook the web extensively when the female reacted to their presence aggressively. Upon physical contact, males of *L. tredecimguttatus* begin to lay silk on the female, a behavior termed mate binding or bridal veil, which occurs in several spider families, e.g., *Xysticus*, Thomisidae [75], *Metellina*, Tetragnathidae [76], *Pisaurina*, Pisauridae [77], *Schizocosa*, Lycosidae [78], *Nephila*, Nephilidae [79], *Caerostris*, Araneidae [80] and *L. hesperus*, Theridiidae [81]. In the only experimental study, mate binding in *Nephila pilipes* was shown to be both a physical and chemical signal to the female, and reduces the likelihood of sexual cannibalism [79]. Finally, males engage in oral sexual contact, where they routinely approach the female genital openings with their mouth and deposit saliva into the openings. This behavior was described and quantified in the bark spider *C. darwini* [80], and was briefly mentioned in *L. hasselti* [82]. As in *C. darwini*, oral sexual contact in *L. tredecimguttatus* seems an obligatory part of the male courting repertoire. The significance of this behavior is yet unknown but is suggested to function as a signal to the female and/or providing males with an advantage in sperm competition [80]. Mating behaviors related to sexual conflict (e.g., mate binding, web shaking, web reduction) and their frequency affect mating success in several spider species [72,74,79]. Our lack of support for the correlation between courting activity and mating success in *L. tredecimguttatus* must be interpreted cautiously, as our experimental design has put a large emphasis on male–male competition.

## 5. Conclusions

Our study shows that individual variation in aggression levels plays no direct role in the mating behaviors of the sexually size dimorphic Mediterranean black widow. Instead, male body length affects success in contests over a female and securing mating opportunities. Sexual cannibalism positively relates to a longer insertion in duration. Furthermore, the smaller the ratio between male and female body length, the more likely a female attacked and cannibalized a mate. The male courting repertoire includes web reduction, silk deposition, mate binding and oral sexual contact, all behaviors that likely evolved under sexual conflict.

## Figures and Tables

**Figure 1 biology-10-00189-f001:**
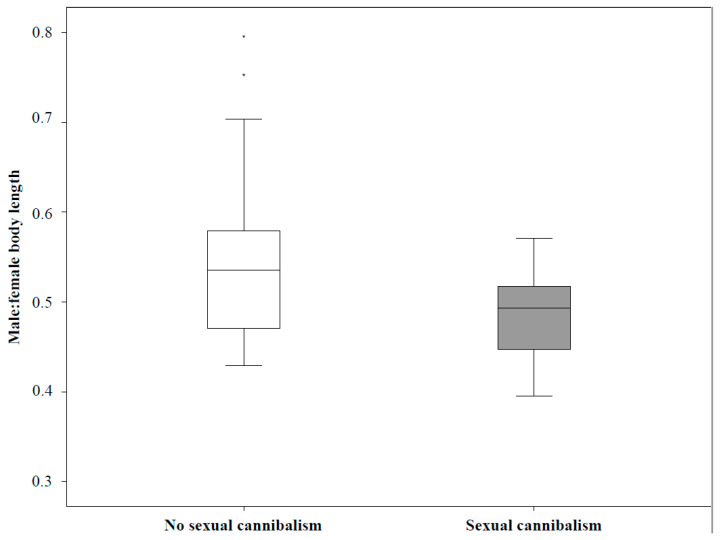
Ratio between male and female body length (male: female) in the mating trials without sexual cannibalism and in the trials where females cannibalized males. * are outliers.

**Figure 2 biology-10-00189-f002:**
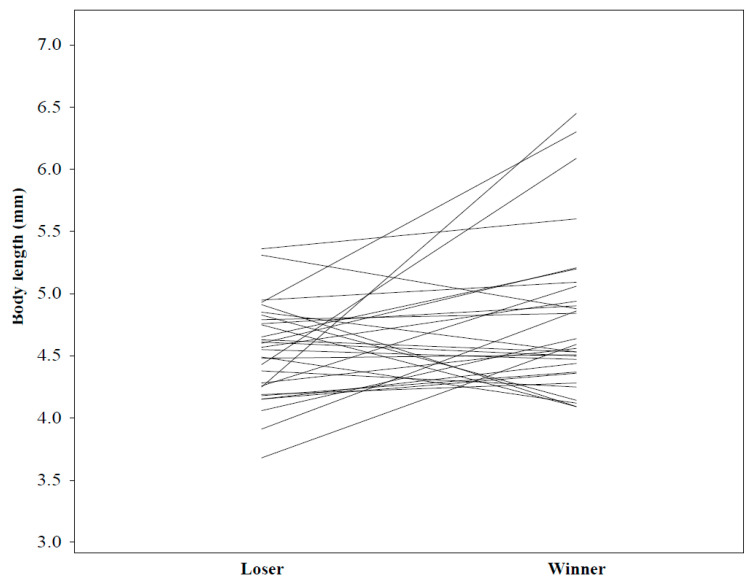
Body lengths of males in male–male competition. Data of a male that lost the contest (left) are lined with his rival that won (right).

**Figure 3 biology-10-00189-f003:**
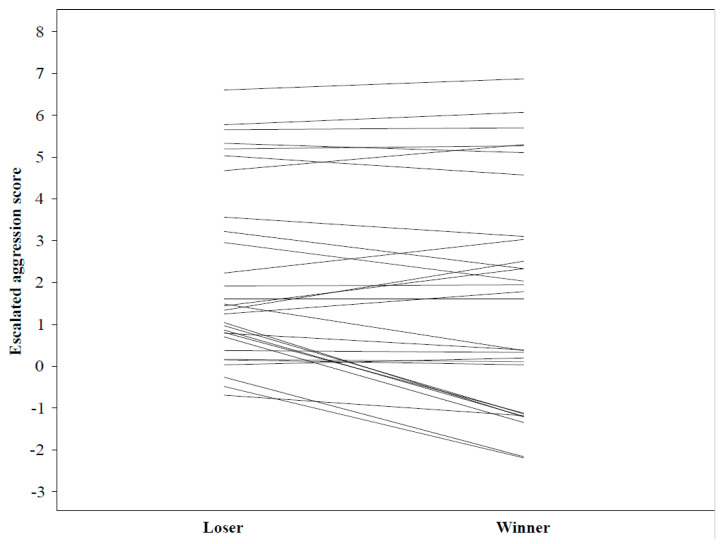
Escalated aggression score of males in male–male competition. Data of a male that lost the contest (left) are lined with his rival that won (right).

**Figure 4 biology-10-00189-f004:**
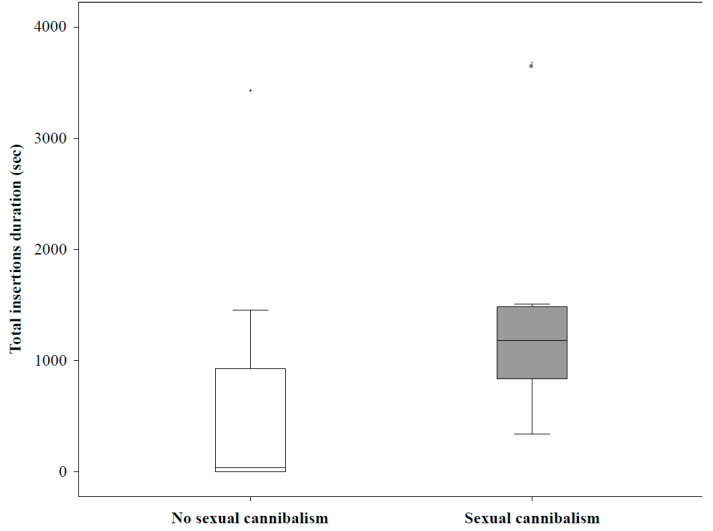
Total copulation duration in the mating trials without sexual cannibalism and in the trials where females cannibalized males. * are outliers.

**Table 1 biology-10-00189-t001:** Repeatability estimates of recorded behaviors in females and males. We ran Markov Chain Monte Carlo (MCMC) glmm in R [36] to partition phenotypic variance in between-individual (Vind) and within-individual variance (Ve), and calculated repeatability as $r\;=\frac{\mathrm{Vind}}{\mathrm{Vind}+\mathrm{Ve}}$ (Nakagawa and Schielzeth 2010).

Behaviors	Vind	Ve	Repeatability
**Females**			
Voracity towards prey	298 [79.74, 535.8]	515.2 [319.9, 737.7]	0.343 [0.109, 0.558]
Prey manipulation	15678 [3379, 30051]	57786 [38679, 78689]	0.168 [0.052, 0.384]
**Males**			
Aggression	0.107 [0.025, 0.215]	0.732 [0.427, 1.059]	0.083 [0.033, 0.289]
Escalated aggression	0.050 [0.020, 0.090]	0.073 [0.039, 0.115]	0.383 [0.206, 0.628]
Attack first	0.536 [<0.001, 1.648]	0.999 [0.999, 0.999]	0.002 [<0.001, 0.452]
Winner-loser	1.640 [<0.001, 3.576]	0.999 [0.999, 0.999]	0.396 [0.075, 0.687]

**Table 2 biology-10-00189-t002:** Pearson’s correlation coefficients among female body size measures (body length, abdomen volume, body condition), voracity towards prey and duration of prey manipulation.

		Abdomen Volume	Body Condition	Voracity twd Prey	Prey Manipulation
Body length	r	0.601	>0.001	−0.128	−0.205
	Sig.	<0.001	1	0.370	0.150
	N	51	51	51	51
Abdomen volume	r	1	0.799	−0.084	−0.126
	Sig.		<0.001	0.556	0.377
	N	51	51	51	51
Body condition	r		1	−0.009	−0.004
	Sig.			0.949	0.977
	N		51	51	51
Voracity twd prey	r			1	0.290
	Sig.				0.033
	N			54	54

**Table 3 biology-10-00189-t003:** The effects of a female’s body size measures, voracity towards prey and duration of prey manipulation on mating success (yes/no), number of copulations and total copulations duration.

	B	Bias	Std. Error	Sig.	Lower 95% CI	Upper 95% CI
**Mating Success**						
Body length	0.021	1.117	49.196	0.973	−6.251	3.493
Abdomen volume	0.010	1.327	37.726	0.666	−0.034	0.229
Voracity twd prey	−0.018	17.178	400.724	0.973	−3.324	8.102
Prey manipulation	−0.003	−0.050	1.230	0.140	−0.021	0.002
Constant	0.518	−147.115	4309.470	0.932	−34.694	46.365
**Number of Copulations**						
(Constant)	0.662	0.115	3.387	0.835	−5.521	7.858
Body length	0.100	−0.018	0.428	0.801	−0.748	0.989
Abdomen volume	0.000	0.001	0.009	0.965	−0.018	0.018
Voracity twd prey	0.480	0.119	0.456	0.299	−0.147	1.622
Prey manipulation	−0.001	1.063 × 10^−0.6^	0.001	0.307	−0.002	0.001
**Total Copulations Duration**						
(Constant)	1.026	2.175	11.873	0.927	−16.882	29.271
Body length	0.581	−0.282	1.460	0.668	−2.792	2.896
Abdomen volume	0.000	0.004	0.027	0.995	−0.045	0.064
Voracity twd prey	0.167	0.172	1.371	0.878	−2.598	3.054
Prey manipulation	−0.004	0.000	0.003	0.184	−0.010	0.002

**Table 4 biology-10-00189-t004:** The effects of a female’s body size measures, voracity towards prey and duration of prey manipulation on female aggression towards males (yes/no) and sexual cannibalism (yes/no).

	B	Bias	Std. Error	Sig.	Lower 95% CI	Upper 95% CI
**Aggression twd Males**						
Body length	0.973	5.219	88.943	0.212	−0.900	5.489
Abdomen volume	−0.012	−0.043	0.969	0.408	−0.084	0.027
Voracity twd prey	−0.322	−3.694	91.801	0.723	−4.404	2.280
Prey manipulation	0.000	0.003	0.063	0.883	−0.005	0.007
Constant	−7.910	−43.821	726.098	0.234	−45.071	7.600
Male:Female body length	−10.674	−1.643	6.651	0.34	−27.795	−1.955
Constant	4.934	0.793	3.443	0.056	0.207	13.799
**Sexual Cannibalism**						
Body length	0.980	−37.828	1228.801	0.233	−1.527	6.383
Abdomen volume	−0.028	−4.479	139.719	0.050	−0.168	0.003
Voracity twd prey	−1.121	−158.807	5020.037	0.289	−8.818	2.149
Prey manipulation	−0.001	−0.284	8.703	0.777	−0.010	0.006
Constant	−5.806	934.841	29547.35	0.411	−47.315	20.136
Male:Female body length	−10.782	−1.572	6.670	0.041	−27.905	−1.464
Constant	4.648	0.755	3.375	0.087	−0.362	13.196

**Table 5 biology-10-00189-t005:** The effects of a male’s body length, escalated aggression and their interaction, on the contest outcome. We run MCMCglmm with experiment ID as a random variable. The results are shown for the full model.

Source	Post-Mean	95% CI	Significance
Intercept	−14.120	[−27.480, −2.267]	0.031
Body length	0.030	[0.004, 0.059]	0.027
Aggression	−0.903	[−11.220, 9.636]	0.913
Body length × Aggression	<0.001	[−0.021, 0.026]	0.984

**Table 6 biology-10-00189-t006:** The effects of a male’s courting activity, body length and escalated aggression on the mating success, number of copulations (insertions), and total duration of copulations.

	B	Bias	Std. Error	Sig.	Lower 95% CI	Upper 95% CI
**Mating Success**						
Court	0.114	0.072	0.773	0.811	−1.026	1.598
Body length	−0.002	0.004	0.019	0.801	−0.019	0.043
Aggression	0.469	0.137	0.756	0.288	−0.692	2.418
Constant	1.681	−1.997	9.205	0.676	−19.046	10.628
**Number of Copulation**						
Court	0.327	0.022	0.199	0.090	−0.028	0.754
Aggression	0.098	0.012	0.168	0.524	−0.227	0.444
Body length	−0.004	0.000	0.003	0.222	−0.009	0.004
(Constant)	2.571	−0.203	1.597	0.084	−1.154	5.053
**Total Duration of Copulations**						
Court	55.184	30.194	188.527	0.755	−263.892	477.369
Aggression	197.369	6.382	145.862	0.154	−77.377	525.638
Body length	−0.331	0.764	3.268	0.891	−4.712	8.702
(Constant)	913.607	−365.070	1554.222	0.466	−3271.467	2931.793

**Table 7 biology-10-00189-t007:** The effects of a male’s courting activity, body length and escalated aggression on the received female aggression and sexual cannibalism.

	B	Bias	Std. Error	Sig.	Lower 95% CI	Upper 95% CI
**Being Attacked by Female**						
Court	−0.102	0.057	5.504	0.849	−1.989	1.498
Aggression	0.438	0.269	2.950	0.304	−0.547	2.251
Body length	−0.011	−0.003	0.071	0.111	−0.033	0.009
(Constant)	4.557	0.818	29.082	0.153	−5.217	14.552
**Being Sexually Cannibalized**						
Court	−0.328	0.600	31.932	0.591	−2.430	1.249
Aggression	0.253	0.939	25.241	0.520	−1.032	1.764
Body length	−0.009	−0.014	0.404	0.181	−0.036	0.011
(Constant)	3.761	4.619	129.730	0.249	−6.113	15.853

## Data Availability

The data presented in this study are openly available at Data Dryad: https://doi.org/10.5061/dryad.bvq83bk7p.

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
