# Peer review of "Body Size, Not Personality, Explains Both Male Mating Success and Sexual Cannibalism in a Widow Spider"

_biology, 2021, doi:10.3390/biology10030189_

Round 1

Reviewer 1 Report

The article focuses on the role of male and female personalities in mating context. The authors conducted two sets of experiments and observed a wide range of behaviours in both sexes. They found out that larger males win male-male contests more often than smaller males and that smaller females are more cannibalistic than larger females.

The study brings interesting results, however, their presentation and discussion can be improved (see below). As I am not perfectly familiar with personality research, I am wondering whether two measures on an individual are sufficient to speak about personalities.

In the Methods, some relevant information concerning the experimental set-up are missing, so it would be good to add them. The same holds for the sample sizes that are shown only in the tables.

To my taste there are too many tables and their interpretation by the reader does not have to be as straight forward as figures. I suggest to put them in the Supplement and rather present the significant results or results most relevant to the hypotheses raised by the authors in form of figures.

Below you can find specific comments.

Line 42 and further in the text: unify throughout the text how you write “aggressive spill-over hypothesis”, sometimes you use quotation marks and hyphen, sometimes not.

Line 68 – 72: This sentence is a bit awkward, rewrite.

Line 73 - 82: A new paragraph for spiders.

Line 77 - 80: …more aggressive males are less likely cannibalized…

Line 111: Mating success in terms of what?

Line 114: Please specify where were the spiders collected (country, area).

Line 119 – 122: How were the experimental females fed? Was food provided once the females built the web in the metal construction (except of the two mealworms per female provided to test the voracity)?

Line 125 – 129: The heading refers to females only, yet you mention the males; I think you can omit the paragraph.

Line 130: How many days/hours were the feeding trial apart? Was that conducted with virgin or mated females? Did the female build a new web for each experiment? When was the last feeding before each of the experiment? Is the species nocturnal? Were these experiments conducted during day or night (under normal or red light)? If the spiders are nocturnal and the trials were conducted during their day time, could be female foraging affected?

Line 144 – 147: Was the male tested twice on the web of the same female or two different females? Were these the same pairs of males in both trials or different males were paired for the second trial?

Line 150: This is the first and last time you mention sample size in the text. I suggest to add sample sizes here in the Methods or into the Results.

Line 152 – 154: You mention that some males lost legs – were these males used again in the second trial? Injured male who does not attack in the second trial does not have to be less aggressive and/or consistent in their behaviour but simply unable to attack - can we then speak about consistency?

Line 161: Define aggressive movement? Does it mean that they were more rapid than the normal approaches?

Line 169 – 171: How was the movement towards opponent different from the attack? Less rapid approach?

Line 184 – 191: Would be good to have sample sizes here (and also in the rest of the Methods, as mentioned above).

Line 197 – 199: For a reader unfamiliar with the mating procedure in widow spiders it would be good to describe it first or at least explain some terms (e.g. web destruction). Does web destruction involve also silk deposition?

Line 227 – 228: Could you calculate a body condition from your measurements? It is often used in studies with spiders incl. widow spiders to estimate fitness (e.g. Jakob EM, Marshall SW, Uetz GW, 1996. Estimating fitness: a comparison of body condition indices. Oikos 77: 61-67; Andrade, MC (2003). Risky mate search and male self-sacrifice in redback spiders. Behavioral Ecology14(4), 531-538 and many others) and it may serve well here too.

Line 262 – 263: You introduce the term ‘escalated aggression’ without any explanation. Please add.

Line 351 – 352: Have you tried to analyse the effect of the relative size in the pair (male:female size)?

Line 380: I suggest ‘insertion’ instead of ‘copulation’ (also for the line 391).

Line 395 – 397 and 425 – 429: If smaller females kill males to compensate for their sizes, wouldn’t you expect the same for attacking the prey? Especially in this species, where males are considerably smaller than the females and likely do not represent valuable prey item. Could it be that smaller females are more choosy (if cannibalism reflects female choosiness)? E.g. due to fact that their resources are more limited than those of larger females?

Line 458 – 497: Consider shortening this part and moving it to Methods or Results.

Line 467 – 473: I suggest to remove this part.

Line 478: …begin to lay silk on the female…

Line 483: Thomisidae

Author Response

REVIEWER 1:

Line 42 and further in the text: unify throughout the text how you write “aggressive spill-over hypothesis”, sometimes you use quotation marks and hyphen, sometimes not.

Corrected, we now use “aggressive spillover hypothesis” throughout. Lines 15, 20, 52 etc.

Line 68 – 72: This sentence is a bit awkward, rewrite.

Corrected. We have rephrased and split the long sentence into two.

“For example, in great tits (Parus major) and zebra finches (Taeniopygia guttata), behavioural traits are involved in mate choices of both sexes. Similarly, older males of the banded wren (Thryothorus pleurostictus) sing more consistent songs and are more likely to mate than younger males (de Kort et al., 2009).” Lines 72-76.

Line 73 - 82: A new paragraph for spiders.

Paragraph added. Line 79.

Line 77 - 80: …more aggressive males are less likely cannibalized…

Corrected. Lines 84-85.

Line 111: Mating success in terms of what?

Explanation added.

“In females, large body size advertises high fecundity (Head, 1995; Prenter et al., 1999), and we thus predicted that larger females are more attractive to males and achieve higher mating success (number and duration of copulations).” Lines 118-120.

Line 114: Please specify where were the spiders collected (country, area).

Locality including coordinates added. Lines 131-132.

Line 119 – 122: How were the experimental females fed? Was food provided once the females built the web in the metal construction (except of the two mealworms per female provided to test the voracity)?

Yes, the feeding and spraying regimes remained the same. We have now added this explanation.

“To facilitate web construction for later experiments, we moved penultimate females onto wooden or metal construction (20 x 12 x 11 cm), consisting of a horizontal plate and six vertical sticks (Supplementary material). The feeding and spraying regimes remained the same for spiders on these constructions.” Lines 140-142.

Line 125 – 129: The heading refers to females only, yet you mention the males; I think you can omit the paragraph.

Thank you for noticing this. We now omit parts of the text and move the rest to the end of section 2.1 Study animals, were it was originally intended. Lines 144-146.

“The feeding and spraying regimes remained the same for spiders on these constructions. We recorded each experiment using a Panasonic HC-V180 camcorder, and we conducted all behavioural quantifications from recordings.” Lines 142-148.

Line 130: How many days/hours were the feeding trial apart? Was that conducted with virgin or mated females? Did the female build a new web for each experiment? When was the last feeding before each of the experiment? Is the species nocturnal? Were these experiments conducted during day or night (under normal or red light)? If the spiders are nocturnal and the trials were conducted during their day time, could be female foraging affected?

We added additional explanation about experiments. We address the timing of experiments, female mating status, and the feeding regime, but not web building. Namely, cobweb spiders do not rebuild webs daily as do orb weavers, but rather reside in their relatively permanent webs that they occasionally adjust.

“We quantified female voracity as her reaction to prey. Each female (N = 54) was tested for voracity twice before being in the mating experiment. After reaching adulthood, all females were fed with one food item, three days later tested for voracity and allowed to eat the prey, and again then tested for voracity three days later, then finally put into a mating trial after another three–day period. Before each experiment, we set the female’s web in front of a black background and between lights, to visualize the web and spider for video recording. These spiders are active during the day and night, so the lights likely did not affect results.” Lines 151-158.

Line 144 – 147: Was the male tested twice on the web of the same female or two different females? Were these the same pairs of males in both trials or different males were paired for the second trial?

Additional explanations added.

“We quantified a male’s aggression as his behaviours towards a conspecific male. Each male (N = 59) was tested twice, once on an empty adult female web and once in a mating trial. Female webs were chosen at random. In the first experiments, we randomly selected two males and placed them on an empty female web (Supplementary material). Males were paired with a different opponent in each test. Because most males matured within 1-2 weeks of each other, male age was ignored when pairing for personality experiments.” Lines 170-175.

Line 150, 184 – 191: This is the first and last time you mention sample size in the text. I suggest to add sample sizes here in the Methods or into the Results.

Thank you for noticing. We have added sample sizes. Lines 151, 171, 217…

Line 152 – 154: You mention that some males lost legs – were these males used again in the second trial? Injured male who does not attack in the second trial does not have to be less aggressive and/or consistent in their behaviour but simply unable to attack - can we then speak about consistency?

Yes, these males were reused as the loss of a leg did not appear to affect their behaviour. We added explanations to the text.

“Because males were highly aggressive during the first contest, several losing a leg, we conducted the second contest as part of the mating trial to prevent larger injuries before mating trials.” Lines 181-183.

…and later

“Males injured in the first experiments were not disqualified from mating experiments, because loss of a leg did not appear to hinder their behaviour.” Lines 189-190.

Line 161: Define aggressive movement? Does it mean that they were more rapid than the normal approaches?

Line 169 – 171: How was the movement towards opponent different from the attack? Less rapid approach?

We rephrase and now write rapid instead of aggressive for attack. While this definition seems arbitrary, it is easily scored, and visible in our supporting video material.

“We defined an attack as any rapid movement towards the opponent in proximity (see Supplementary Materials). ” Lines 193-194.

“Number of movements towards opponent: total number of times that a male responded to the presence of its opponent by stopping his courting behaviour and walking towards the opponent, occasionally shaking the web. Chance encounters were not included, and were defined as random encounters between two courting males.” Lines 201-204.

Line 197 – 199: For a reader unfamiliar with the mating procedure in widow spiders it would be good to describe it first or at least explain some terms (e.g. web destruction). Does web destruction involve also silk deposition?

As per request of reviewer #2, we now term this behaviour “web reduction”, and expand the description.

“Web reduction: a male collapsing some of the female’s web using his chelicerae.” Lines 232-233.

Line 227 – 228: Could you calculate a body condition from your measurements? It is often used in studies with spiders incl. widow spiders to estimate fitness (e.g. Jakob EM, Marshall SW, Uetz GW, 1996. Estimating fitness: a comparison of body condition indices. Oikos 77: 61-67; Andrade, MC (2003). Risky mate search and male self-sacrifice in redback spiders. Behavioral Ecology14(4), 531-538 and many others) and it may serve well here too.

We calculated body condition and used it in the correlational analyses. Lines 274-275, 354, Table 2.

However, we could not include body condition to the regression analyses due to multicollinearity, as it logically correlates both to abdomen volume as a proxy of mass, and with body size. We nevertheless ran the analyses to explore the results. Body condition however did not influence mating success, etc.

Line 262 – 263: You introduce the term ‘escalated aggression’ without any explanation. Please add.

We indeed define escalated aggression only later in the text (section 3.2). We added an explanation here, too.

“Next, we tested for the relationships between body size measures (body length, abdomen volume and body condition) and behaviours that were significantly repeatable, i.e. voracity towards prey (primary factor from PCA analysis, see Results) and duration of prey manipulation in females, and with escalated aggression (aggression that continue beyond just a confrontation, e.g. fighting; also see 3.2) and winner-loser in males.” Lines 307-311.

Line 351 – 352: Have you tried to analyse the effect of the relative size in the pair (male:female size)?

Our aim was to analyse data according to our predictions and we do not have much freedom to add novel variables into the existing models, because our sample size in mating is 30 and we need to avoid multicollinearity. However, we decided to add the test on how relative size in the pair affects the probability of female attack and sexual cannibalism (mate dimorphism explanation for SC). We also tested the relationship between insertion duration, and female aggression and sexual cannibalism. Lines 120-127, 320-323, 387-390, Table 4, Figure 1, Figure 4.

Line 380: I suggest ‘insertion’ instead of ‘copulation’ (also for the line 391).

We agree have made the changes. Lines 440, 445.

Line 395 – 397 and 425 – 429: If smaller females kill males to compensate for their sizes, wouldn’t you expect the same for attacking the prey? Especially in this species, where males are considerably smaller than the females and likely do not represent valuable prey item. Could it be that smaller females are more choosy (if cannibalism reflects female choosiness)? E.g. due to fact that their resources are more limited than those of larger females?

We agree omitted the sentence. Otherwise, while being small in body mass, eating a conspecific might have other physiological benefits. (Wilder et al. 2009). Wilder, S. M., Rypstra, A. L. and Elgar, M. A. (2009). The importance of ecological and phylogenetic conditions for the occurrence and frequency of sexual cannibalism. Annu. Rev. Ecol. Evol. Syst. 40, 21–39

Line 458 – 497: Consider shortening this part and moving it to Methods or Results.

As a secondary goal in this study was to quantify mating behaviours connected to sexual size dimorphism, we use this section to discuss our results. For most of these extreme behaviours it is unclear whether they are causes or consequences of SSD, and quantifications are very rare. Because reviewer 2 did not find it excessive, we decided to leave it in the present form.

Line 467 – 473: I suggest to remove this part.

Similar to the above comment, we agree with reviewer 2 that this is valuable data in light of the evolution of extreme sexual behaviours connected to SSD. We added explaining text to methods and rephrased the discussion part.

“After noticing web reduction and silk deposition behaviour by males, we conducted a subsequent experiment, testing for pheromones in female silk. We collected entire female webs on a sterile inoculation loop. Following Scott et al. (2018b), we then washed the collected silk in methanol to extract potential pheromones. We soaked sterile filter paper in 50 µl of silk extract, and put each filter paper on a T-shaped climbing structure (Scott et al. 2018). We covered each filter paper with a plastic cage to prevent males from contacting the it. We presented 10 males each to their own such silk extract and tested whether they will locate the filter paper and start courtship.” Lines 256-263.

“All males (N = 10) in the subsequent pheromone trial, located the filter paper despite prevented from direct contact, and started depositing their own silk onto the plastic cage.” Lines 425-427.

“Both male silk deposition and web reduction in L. hesperus reduce the attractiveness of the female web for other males (Scott et al., 2015). As our pheromone trials indicate the presence of airborne pheromones in female silk of L. tredecimguttatus, male web reduction and silk deposition in this species suggests a similar function, i.e. masking the female web to make it less attractive for rival males.” Lines 546-566.

Line 478: …begin to lay silk on the female…

Changed accordingly. Lines 578-579.

Line 483: Thomisidae

Thank you for noticing, corrected. Line 583.

Reviewer 2 Report

This paper investigates relationships between male aggression, female voracity, and mating outcomes including sexual cannibalism, providing very interesting data that does not support the aggressive spillover hypothesis. I enjoyed reading the paper and I think it makes a valuable contribution to the literature on sexual cannibalism as well as important data about the sexual behaviour of widow spiders. I have only minor suggestions for improvement of the manuscript.

I will note to the editors that as a reviewer it is very annoying to receive the manuscript without line numbers, as this makes it difficult to point out to the authors where specific changes should be made.

The title doesn’t seem to capture the most interesting results, in my opinion. I suggest something like “Body size, not personality, explains both male mating success and female cannibalism in a widow spider” or “Larger--but not more aggressive--males have increased mating success…”

I will note that I am not an expert on animal personality reaserch methods, so I cannot comment on the specific statistical methods but in general the approaches taken seem reasonable. I would appreciate seeing some figures illustrating relationships between variables like male size and contest/copulation success and female mass and cannibalism frequency, in addition to the tables.

The dryad link does not yet seem to be active yet so I was not able to access the supplementary materials, but having these videos available will be valuable.

Page 3 (final sentence of introduction)

“large size advertises high fecundity” – by size here do you mean mass or body size, or both? Please be careful throughout the paper to specify what aspect of size you are referring to, especially when discussing foraging success before and after maturity. Body size (measured as leg length or cephalothorax width) will be fixed at maturity, but mass (condition) can fluctuate dramatically with feeding.

Methods:

Please specify where spiders were collected from (i.e., country, lat/long)

How old were males and females when they were used in trials? If some females were tested earlier and others later after maturity, this could influence their mass, since an older female would have been fed more times. Related to this, I wonder if lighter females with smaller abdomens are simply hungrier, and this explains their higher rate of cannibalism (as in redbacks; see Andrade 1998 “Female hunger can explain variation in cannibalistic behavior despite male sacrifice in redback spiders”). I’ve found with western widows that even females of the exact same age who have been offered the exact same number of prey items can vary dramatically in mass because some of them just aren’t as good at capturing all the prey they are given.

Section 2.5

Please add more detail about “body length” is this total body length, or length of a limb or cephalothorax? If total body length, it is not surprising that it is strongly correlated with abdomen volume, since the abdomen can shrink and expand. It’s not clear to me how much information total body length would add in the models since it is kind of a composite of size at maturity plus condition (which is already captured by abdomen volume), rather than a measure of fixed size at maturity.

3.4 Can you add some information about the timing of sexual cannibalism? Were the 25% of males that were cannibalized during the first copulation still transferring sperm while being cannibalized? Did the female always end the copulation by attacking the male, or did the female sometimes not attack until he was trying to remove his pedipalp? Similarly, were the males who copulated twice cannibalized during or after the second insertion? It would also be interesting, if you have the data, to compare the duration of copulations for males that were and were not cannibalized.

Very cool about the oral sexual contact! This definitely happens in Latrodectus hesperus too and I wonder if it is a “preadaptation” for immature mating? I still have not seen how the males actually chew through the exoskeleton to access the genitalia but maybe the chewing action is similar and the saliva helps somehow?

4.1 End of first paragraph and end of 4th paragraph. I am skeptical about the argument that lighter, more cannibalistic females matured at smaller size without data on their size at maturity (e.g. leg length or cephalothorax width) that can be separated from abdomen size, which could reflect foraging success as an adult or subadult, depending on the females’ ages during the experiments. I think it definitely makes sense that hungrier females should be more cannibalistic, to increase their fecundity, but that it could just as easily be to compensate for poor foraging success so far as an adult as during development.

4.2 New data (that are interesting and valuable) about silk pheromones are presented here that were not mentioned in the methods or results. I think if the authors want to report on this they should be mentioned earlier. Please explain the bioassay and what male behaviour was measured (movement toward the filter paper?) to confirm airborne vs. contact pheromones.

Final paragraph: please change “Female size” to “abdomen size” or “mass” (or hunger!) to more accurately reflect the relationship between size and cannibalism that is reported here  

Specific small comments  

Abstract: change “lightweight” to “lighter”

Page 2:

Change “or show even the opposite as predicted” to “or studies find effects in the opposite direction than predicted”

Throughout (or at least once) please consider using the term “web reduction” instead of “web destruction” since there are several other papers describing this behaviour in widows and other spider species which use “web reduction” and this will make it easier for readers searching for information about this behaviour in widows and other spiders

Throughout: consider saying “cannibalized” or “consumed” rather than “devoured” which has negative connotations (see the paper “Sexual stereotypes: the case of sexual cannibalism” https://doi.org/10.1016/j.anbehav.2012.12.008)

Section 2.3

#1 how did you define what was an “aggressive” movement toward the opponent. Any rapid, directed movement toward the opponent, perhaps? Aggressive is subjective.

#5 how did you define a chance encounter? Again a bit subjective.

Section 2.4

#9 define “aggression” toward the male. What actual behaviours were counted?

Table 1: change “looser” to “loser”

Table 2: change “>0.001” to “<0.001”

3.4 “perceived female aggression” perceived by the male or by the researchers?

Page 11

Change “opposite of the proposed…” to “opposite to the outcome proposed by the aggressive…”

Page 12: change “Thpmisidae” to “Thomisidae”

Page 13: change “evolved sexual conflict” to “evolved under sexual conflict”

Thanks for the opportunity to review this very nice manuscript!

Sincerely,

Catherine Scott

Author Response

REVIEWER 2

This paper investigates relationships between male aggression, female voracity, and mating outcomes including sexual cannibalism, providing very interesting data that does not support the aggressive spillover hypothesis. I enjoyed reading the paper and I think it makes a valuable contribution to the literature on sexual cannibalism as well as important data about the sexual behaviour of widow spiders. I have only minor suggestions for improvement of the manuscript.

We thank the reviewer for this evaluation.

The title doesn’t seem to capture the most interesting results, in my opinion. I suggest something like “Body size, not personality, explains both male mating success and female cannibalism in a widow spider” or “Larger--but not more aggressive--males have increased mating success…”

Thank you for these suggestions. We agree and changed the title to “Body size, not personality, explains both male mating success and sexual cannibalism in a widow spider”. Lines 2-4.

I will note that I am not an expert on animal personality reaserch methods, so I cannot comment on the specific statistical methods but in general the approaches taken seem reasonable. I would appreciate seeing some figures illustrating relationships between variables like male size and contest/copulation success and female mass and cannibalism frequency, in addition to the tables.

This is great idea. We now included four figures.

The dryad link does not yet seem to be active yet so I was not able to access the supplementary materials, but having these videos available will be valuable.

Instead of the sharing link for preliminary data, we have by mistake supplied the doi that will work only upon publications. We have now corrected this. Lines 620-621.

Page 3 (final sentence of introduction)

“large size advertises high fecundity” – by size here do you mean mass or body size, or both? Please be careful throughout the paper to specify what aspect of size you are referring to, especially when discussing foraging success before and after maturity. Body size (measured as leg length or cephalothorax width) will be fixed at maturity, but mass (condition) can fluctuate dramatically with feeding.

Well, body size and body mass are usually correlated in spiders, also in the present study (Table 1). We revised the sentence.

“In females, large body size advertises high fecundity (Head, 1995; Prenter et al., 1999), and we thus predicted that larger females are more attractive to males and achieve higher mating success (number and duration of copulations). ” Lines 118-120.

Methods:

Please specify where spiders were collected from (i.e., country, lat/long)

Locality and coordinates added. Lines 131-132.

How old were males and females when they were used in trials? If some females were tested earlier and others later after maturity, this could influence their mass, since an older female would have been fed more times. Related to this, I wonder if lighter females with smaller abdomens are simply hungrier, and this explains their higher rate of cannibalism (as in redbacks; see Andrade 1998 “Female hunger can explain variation in cannibalistic behavior despite male sacrifice in redback spiders”). I’ve found with western widows that even females of the exact same age who have been offered the exact same number of prey items can vary dramatically in mass because some of them just aren’t as good at capturing all the prey they are given.

We had a rigorous protocol to control for time after maturation. We rephrased this text to clarify.

“After reaching adulthood, all females were fed with one food item, three days later tested for voracity and allowed to eat the prey, and again then tested for voracity three days later, then finally put into a mating trial after another three–day period. Lines 152-155.

Section 2.5

Please add more detail about “body length” is this total body length, or length of a limb or cephalothorax? If total body length, it is not surprising that it is strongly correlated with abdomen volume, since the abdomen can shrink and expand. It’s not clear to me how much information total body length would add in the models since it is kind of a composite of size at maturity plus condition (which is already captured by abdomen volume), rather than a measure of fixed size at maturity.

The total body length was measured as prosoma + opisthosoma, the measure which is commonly used in spider biology (as described in Cheng & Kuntner 2014). However, we also measured only the abdomen length as part of the formula we used to calculate abdomen volume. In addition, we calculated body size condition (see above). Lines 265, 274-275.

Cheng, R. C., & Kuntner, M. (2014). Phylogeny suggests nondirectional and isometric evolution of sexual size dimorphism in argiopine spiders. Evolution, 68(10), 2861-2872.

3.4 Can you add some information about the timing of sexual cannibalism? Were the 25% of males that were cannibalized during the first copulation still transferring sperm while being cannibalized? Did the female always end the copulation by attacking the male, or did the female sometimes not attack until he was trying to remove his pedipalp? Similarly, were the males who copulated twice cannibalized during or after the second insertion? It would also be interesting, if you have the data, to compare the duration of copulations for males that were and were not cannibalized.

Indeed, all cases of sexual cannibalism occurred to terminate male palpal insertions. We add this explanation. We also added the requested analyses. Female aggression and sexual cannibalism were more likely when total copulation duration were longer, but the relationships were not significant.

“Females were aggressive in 60 % of mating experiments, and sexual cannibalism occurred in 45 % of experiments. In all experiments with sexual cannibalism, females used it to terminate mating. All cases of cannibalism occurred during palpal insertion, and the sexual organs of males remained lodged inside female genitals even while being consumed. Females attacked males more often during first insertion (40 % of experiments). Female aggression and sexual cannibalism were more likely when total copulation duration were longer, however the relationships were not significant (aggression towards a mate, B=0.001, SE=0,001, Wald=2.575, p=0.109; sexual cannibalism, B=0.001, SE=0,001, Wald=3.455, p=0.063; Figure 4).” Lines 445-454.

Very cool about the oral sexual contact! This definitely happens in Latrodectus hesperus too and I wonder if it is a “preadaptation” for immature mating? I still have not seen how the males actually chew through the exoskeleton to access the genitalia but maybe the chewing action is similar and the saliva helps somehow?

We agree that it would be interesting to explore the evolution of oral sexual contact and immature mating. Currently, we can only hypothesize about the function of oral sexual contact, and we have not tested for immature mating in our experiments, but both could well be evolutionary consequences of similar selection pressures.

4.1 End of first paragraph and end of 4th paragraph. I am skeptical about the argument that lighter, more cannibalistic females matured at smaller size without data on their size at maturity (e.g. leg length or cephalothorax width) that can be separated from abdomen size, which could reflect foraging success as an adult or subadult, depending on the females’ ages during the experiments. I think it definitely makes sense that hungrier females should be more cannibalistic, to increase their fecundity, but that it could just as easily be to compensate for poor foraging success so far as an adult as during development.

We omit the sentence.

4.2 New data (that are interesting and valuable) about silk pheromones are presented here that were not mentioned in the methods or results. I think if the authors want to report on this they should be mentioned earlier. Please explain the bioassay and what male behaviour was measured (movement toward the filter paper?) to confirm airborne vs. contact pheromones.

This was indeed a subsequent experiment, conducted after observing male web reduction and silk deposition. We now added text to the methods and rephrase the discussion slightly.

“After noticing web reduction and silk deposition behaviour by males, we conducted a subsequent experiment, testing for pheromones in female silk. We collected entire female webs on a sterile inoculation loop. Following Scott et al. (2018b), we then washed the collected silk in methanol to extract potential pheromones. We soaked sterile filter paper in 50 µl of silk extract, and put each filter paper on a T-shaped climbing structure (Scott et al. 2018). We covered each filter paper with a plastic cage to prevent males from contacting the it. We presented 10 males each to their own such silk extract and tested whether they will locate the filter paper and start courtship.” Lines 256-263.

“All males (N = 10) in the subsequent pheromone trial, located the filter paper despite prevented from direct contact, and started depositing their own silk onto the plastic cage.” Lines 425-427.

“Both male silk deposition and web reduction in L. hesperus reduce the attractiveness of the female web for other males (Scott et al., 2015). As our pheromone trials indicate the presence of airborne pheromones in female silk of L. tredecimguttatus, male web reduction and silk deposition in this species suggests a similar function, i.e. masking the female web to make it less attractive for rival males.” Lines 564-573.

Final paragraph: please change “Female size” to “abdomen size” or “mass” (or hunger!) to more accurately reflect the relationship between size and cannibalism that is reported here  

Corrected.

“Instead, male body length affects success in contests over a female and securing mating opportunities Sexual cannibalism positively relates to longer copulations duration. Furthermore, the smaller the ratio between male:female body length the more likely a female attacked and cannibalized a mate. “ Lines 601-606.

Specific small comments  

Abstract: change “lightweight” to “lighter”

This part has changed.

Page 2:

Change “or show even the opposite as predicted” to “or studies find effects in the opposite direction than predicted”

Requested change made. Lines 57-58.

Throughout (or at least once) please consider using the term “web reduction” instead of “web destruction” since there are several other papers describing this behaviour in widows and other spider species which use “web reduction” and this will make it easier for readers searching for information about this behaviour in widows and other spiders

Throughout: consider saying “cannibalized” or “consumed” rather than “devoured” which has negative connotations (see the paper “Sexual stereotypes: the case of sexual cannibalism” https://doi.org/10.1016/j.anbehav.2012.12.008)

Thank you for pointing out inconsistencies in terminology: we changed it to web reduction throughout. We also changed devoured to consumed.

Section 2.3

#1 how did you define what was an “aggressive” movement toward the opponent. Any rapid, directed movement toward the opponent, perhaps? Aggressive is subjective.

#5 how did you define a chance encounter? Again a bit subjective.

We rephrase and now write rapid instead of aggressive for attack. While this definition seems arbitrary, it is easily scored, and visible in our supporting video material (hopefully now accessible).

“We defined an attack as any rapid movement towards the opponent in proximity (see Supplementary Materials).” Lines 193-194.

“Number of movements towards opponent: total number of times that a male responded to the presence of its opponent by stopping his courting behaviour and walking towards the opponent, occasionally shaking the web. Chance encounters were not included, and were defined as random encounters between two courting males.” Lines 201-204.

Section 2.4

#9 define “aggression” toward the male. What actual behaviours were counted?

We added a better explanation with examples.

“9.       Sexual aggression: measured binary (yes/no) for whether the female attempted to bite the male or grasped at him with her legs, attempted to cover him in silk).” Lines 245-247

Table 1: change “looser” to “loser”

Table 2: change “>0.001” to “<0.001”

Changed both errors.

3.4 “perceived female aggression” perceived by the male or by the researchers?

Rephrased.

Page 11: Change “opposite of the proposed…” to “opposite to the outcome proposed by the aggressive…”

Page 12: change “Thpmisidae” to “Thomisidae”

Page 13: change “evolved sexual conflict” to “evolved under sexual conflict”

Thank you for noticing. Corrected.

Round 2

Reviewer 1 Report

The authors addressed all my comments and made appropriate changes. I have no further recommendations or comments.